# Low Energy Availability in Athletes 2020: An Updated Narrative Review of Prevalence, Risk, Within-Day Energy Balance, Knowledge, and Impact on Sports Performance

**DOI:** 10.3390/nu12030835

**Published:** 2020-03-20

**Authors:** Danielle M. Logue, Sharon M. Madigan, Anna Melin, Eamonn Delahunt, Mirjam Heinen, Sarah-Jane Mc Donnell, Clare A. Corish

**Affiliations:** 1School of Public Health, Physiotherapy and Sports Science, University College Dublin, Belfield, V04 V1W8 Dublin 4, Ireland; 2Sport Ireland Institute, Sports Campus Ireland, Abbotstown, D15 PNON Dublin, Ireland; smadigan@instituteofsport.ie (S.M.M.); sjmcdonnell@instituteofsport.ie (S.-J.M.D.); 3Department of Sports Science, Linnaeus University, 392 34 Kalmar, Sweden; anna.melin@lnu.se; 4School of Public Health, Physiotherapy and Sports Science and Institute for Sport and Health, University College Dublin, V04 V1W8 Dublin, Ireland; eamonn.delahunt@ucd.ie; 5Assistant Professor Mirjam Heinen, School of Public Health, Physiotherapy and Sports Science, University College Dublin, V04 V1W8 Dublin, Ireland; mirjam.heinen@ucd.ie; 6Associate Professor Clare Corish, School of Public Health, Physiotherapy and Sports Science, University College Dublin, V04 V1W8 Dublin, Ireland; clare.corish@ucd.ie

**Keywords:** low energy availability, relative energy deficiency in sport, health and performance

## Abstract

Low energy availability (EA) underpins the female and male athlete triad and relative energy deficiency in sport (RED-S). The condition arises when insufficient calories are consumed to support exercise energy expenditure, resulting in compromised physiological processes, such as menstrual irregularities in active females. The health concerns associated with longstanding low EA include menstrual/libido, gastrointestinal and cardiovascular dysfunction and compromised bone health, all of which can contribute to impaired sporting performance. This narrative review provides an update of our previous review on the prevalence and risk of low EA, within-day energy deficiency, and the potential impact of low EA on performance. The methods to assess EA remain a challenge and contribute to the methodological difficulties in identifying “true” low EA. Screening female athletic groups using a validated screening tool such as the Low Energy Availability in Females Questionnaire (LEAF-Q) has shown promise in identifying endurance athletes at risk of low EA. Knowledge of RED-S and its potential implications for performance is low among coaches and athletes alike. Development of sport and gender-specific screening tools to identify adolescent and senior athletes in different sports at risk of RED-S is warranted. Education initiatives are required to raise awareness among coaches and athletes of the importance of appropriate dietary strategies to ensure that sufficient calories are consumed to support training.

## 1. Introduction

Considerable research has been undertaken to understand the health and performance consequences of relative energy deficiency in sports (RED-S), a condition frequently observed among high performing male and female athletes [1,2]. It is widely acknowledged that low energy availability (EA), described as inadequate energy intake relative to exercise energy expenditure, is the main factor triggering the unfavourable health and performance consequences associated with RED-S [3]. Since the publication of the International Olympic Committee (IOC) consensus papers on RED-S in 2014 and 2018 [1,2], scientific evidence for the risk of and performance consequences of low EA has grown [4,5].

Although an understanding of the physiological effects of low EA in male athletes remains limited, a recent cross-sectional study conducted on males participating in endurance training reported that regular exposure to high levels of chronic intense and long duration training is associated with a decreased libido score. [6]. Therefore, the IOC and other researchers in the field [7] advise that research should now focus on the energy demands and performance criteria of males engaged in a range of sports as the majority of research in the past has focused on women [2].

Following publication of our 2018 review on the prevalence of low EA and associated health and performance consequences [7], the volume of research on RED-S has substantially increased (peer-reviewed articles in 2018 (*n* = 21), 2019 (*n* = 24) vs. 2016 (*n* = 6) and 2017 (*n* = 10)). The purpose of this investigation is to identify other potential methods for assessing low EA given that the current methods used remain challenging. Furthermore, the review highlights recent data on the prevalence and risk of low EA, within-day energy deficiency and the health and sporting performance consequences associated with low EA. Furthermore, the published literature on awareness and knowledge of RED-S among athletes and coaches has been evaluated to identify gaps in the practical application of low EA education in sports [8,9,10,11].

## 2. Methodology

This is an update of a narrative review [7] which was conducted using targeted database searches, for example, PubMed, Google Scholar and Web of Science. Combinations of the following key search terms were included: athlete, EA, low EA, low EA risk, within-day energy balance, low EA knowledge and awareness of low EA, nutrition education/diet intervention and RED-S. Articles published between 2017 and 2019 were considered if they were published in English, available in full text and were conducted among trained or exercising human subjects. The inclusion criteria was as follows: only studies that quantified EA by assessing energy intake, exercise energy expenditure (EEE) and body composition and that investigated symptoms associated with low EA within the text of the manuscript were included in this review. The reference lists of retrieved articles were also reviewed to identify any articles not identified by the database searches. Animal studies were excluded. The quality and strength of the supporting evidence were graded according to the criteria of the Scottish Intercollegiate Guidelines Network (SIGN) [12]. The SIGN grading criteria include the assessment of study design and its ability to minimise the possibility of bias as well as an evaluation of the methodological quality, quantity, consistency, and applicability of a study’s results to the evidence base.

## 3. Results

### 3.1. Low Energy Availability

Recent studies have investigated the prevalence of low EA in various sports [8,13,14,15,16,17,18,19,20,21,22]. Prevalence ranges from 22% to 58% (Table 1). It is apparent that low EA exists in males as well as females as evidenced by studies conducted in male road cyclists [9] and elite distance athletes [23,24,25]. Nonetheless, accurate estimation of the prevalence of low EA remains problematic due to continuing variability in the methods used to estimate EA (Table 2). Furthermore, there are some limitations to the application of clinically low EA < 30 kcal/kg fat free mass (FFM)/kg/day [3] in cross-sectional studies. Self-reported nutritional data in free-living athletes have failed to find clear thresholds or associations between EA and objective measures of energy conservation or health impairment such as disruption to metabolic hormones [24,26] and menstrual disturbances [27,28,29]. Most recently (Table 3), athletes are being screened for symptoms of low EA using questionnaires that screen for physiological symptoms associated with the Female Athlete Triad (Triad) and RED-S, an example of which is the Low EA in Females Questionnaire (LEAF-Q) [30]. The current evidence implies that clinical, individual assessment of the diagnostic criteria of the Triad [31], the LEAF-Q [30] and the RED-S clinical assessment tool (RED-S CAT) [32] is necessary. This assessment should be conducted with an evaluation of disordered eating behaviours and reproductive biomarkers (blood biochemical assessment of sex-hormones and prospective collection of oestrogen and progesterone during one or more menstrual cycles). This may represent a more objective and accurate indicator of optimal EA for health in females than the estimation of EA using dietary and training logs [24]. Nevertheless, further research is warranted as there is limited research on the effect of hormonal contraception on physiological function and biochemical assessment of low EA/RED-S.

Furthermore, a low ratio between measured and predicted resting metabolic rate (RMR) is an acknowledged marker of low EA (RMR ratio < 0.90) [27]. Associations between suppressed RMR and elevated low EA risk scores in female ballet dancers and with higher training volume in male ballet dancers have recently been observed [38]. These results indicate low RMR ratio as a potential surrogate marker for low EA. However, the prevalence of suppressed RMR ratio using dual-energy X-ray (DXA) and different RMR predictive equations such as the Harris–Benedict and Cunningham equations showed large variability in both male and female professional ballet dancers, ranging from 25% to 80% and 35% to 100% in males and females, respectively [38]. Thus, the identification of athletes at risk of low EA varies greatly depending on the predictive equation used. A recent review of the validity of RMR predictive equations to assess low EA in athletes emphasized that predictive equations that do not have FFM incorporated within the algorithm are unsuitable for use in athletes and that accurate RMR laboratory protocols are essential when monitoring EA [42]. Thus, the development of a reliable RMR laboratory protocol in athletes is needed before measured RMR or the RMR ratio can reliably be used to diagnose low EA [27,38,43].

### 3.2. Low Energy Availability Risk

The risk of low EA has been investigated using surrogate markers or self-reported symptoms of low EA in various athlete populations including elite para-athletes [34], adolescent/young adult [35,36] and Olympic athletes [40], female sprinters [41], male jockeys [39] and recreationally active individuals [33,37], and ranges from 14% to 63% (Table 3). The largest cross-sectional study investigating self-reported health and performance outcomes linked to low EA and RED-S [5] placed emphasis on including body systems beyond reproductive function and bone health such as metabolic, haematological, psychological and cardiovascular health and gastrointestinal function. The authors acknowledged that associations identified in their study (Table 3) were based on self-reported data and highlight the need to investigate the health and performance components of RED-S in a controlled setting, whereby low EA is measured under strict conditions in a laboratory setting, to understand causative pathways [5]. Another cross-sectional study supports associations between low EA risk and self-reported medical illness [40]. Moreover, associations have been described between low EA risk and stress fractures, absence from training for >22 days due to illness and reported adherence to a gluten-free diet [37]. These findings suggest that those at risk of low EA can present with symptoms other than those traditionally expected e.g., menstrual irregularities, and highlight the complexity of identifying individuals at risk [40]. The LEAF-Q enables early recognition of active females at risk of low EA by evaluating the presence of symptoms associated with low EA, such as menstrual and gastrointestinal dysfunction, injury history, as well as oral contraceptive use [30]. The LEAF-Q is an easily administered questionnaire that has been validated for use in athletic endurance-trained females [30]. As such, it offers a validated method to investigate risk of low EA in large, exercising cohorts and alleviates some of the challenges associated with directly measuring EA [44]. In the measurement of the components of the EA equation, significant errors of reliability (e.g., variation in the time and techniques used to estimate EA) and validity (e.g., under/over estimation of EI and/or EEE) can occur [45]. Although low EA risk has been investigated in elite male cyclists [9] using non-validated tools such as a sport-specific questionnaire and clinical interview (SEAQ-I) [9], a screening questionnaire for male athletes similar to the LEAF-Q is required and is currently being developed (the Low Energy Availability in Male’s Questionnaire (LEAM-Q) [2]. Following its development and validation, it is anticipated that potential risk of low EA will be identified in a range of athletic male populations.

#### 3.2.1. Eating Disorders and Exercise Addiction

Disordered eating behaviours and eating disorders are known to occur frequently in elite female athletes, particularly among those competing in weight class or leanness-demanding sports [46]. These conditions are associated with perfectionism as well as compulsive exercise behaviour, coupled with the inability to reduce training load [47] and can lead to unfavourable outcomes such as injury and emotional distress [47]. Recent research describes how compulsive exercise behaviour is associated with symptoms of disordered eating behaviour, perfectionism and obsessive-compulsive characteristics in long-distance runners of both sexes [48]. We encourage further exploration of the relationship between compulsive exercise behaviour and RED-S, since individuals demonstrating compulsive exercise behaviour may be at greater risk of the negative health outcomes associated with low EA [48].

Although most research into RED-S has been conducted in females, a recent study investigated associations between compulsive exercise behaviour, disordered eating symptoms and biomarkers of RED-S in trained male cyclists, triathletes and long-distance runners [49]. Using the Eating Disorder Examination Questionnaire (EDE-Q), higher total exercise dependence scores were associated with disordered eating symptoms. Participants with higher exercise dependence scores trained more frequently (~11 h/week) compared to those with lower scores (~8 h/week). Furthermore, participants with higher scores did not adjust their energy intake to meet increased energy needs, resulting in more pronounced low EA. This was associated with higher cortisol levels compared to those with lower exercise dependence scores [49]. Higher subscale exercise dependence scores were also associated with lower blood glucose, lower testosterone: cortisol ratio and higher cortisol: insulin ratio [49]. This suggests an association between biomarkers of RED-S and compulsive exercise behaviour. Although these findings currently relate only to endurance athletes, determining whether these associations exist in highly-trained athletes competing, in particular, in weight sensitive and team sports, represents an interesting area for future investigation.

Psychological factors such as stress, anxiety and depression can result in or contribute to disordered eating behaviour and low EA in athletes [1]. Moreover, it has been suggested that compulsive exercise behaviour could increase vulnerability to negative health and performance outcomes [48]. Despite the lack of conclusive evidence, mainly due to the absence of studies investigating associations between compulsive exercise, disordered eating behaviour symptoms and biomarkers of RED-S, there appears to be an opportunity to improve athlete health and performance. This may be possible through screening for potential underlying causes of low EA such as compulsive exercise, in conjunction with validated disordered eating behaviour and low EA risk questionnaires. Screening for compulsive exercise should increase awareness of the psychological factors underpinning and contributing to low EA and its associated consequences such as poor mental health. More precisely, identifying the relationship between compulsive exercise, disordered eating behaviour and RED-S in various sports and groups of athletes (e.g., gender, performance level and age) is recommended. Nevertheless, self-reported data need to be interpreted with caution as they may be heavily influenced by athletes’ perceptions, experiences and recollections, thus, further emphasising the need to use well-validated questionnaires to investigate outcomes associated with LEA in the athletic population.

#### 3.2.2. Exercise Hypogonadal Male Condition

The Triad continuum, whereby a female can advance from optimal health status [optimal EA, normal menstrual function and good bone health] to a diseased state [low EA (with and without an eating disorder), menstrual dysfunction and low bone mineral density] at any point throughout her life course, has been extensively investigated over the last 30 years [50,51,52]. Rigorously executed studies have identified low EA, not the stress of exercise, as the underlying mechanism causing disruption to the female hypothalamic-pituitary-gonadal axis [51,53]. Further evidence suggests that endurance exercising males can develop a similar suppression of the reproductive function known as exercise hypogonadal male condition (EHMC) [54] (Figure 1). In EHMC, disruption to the male hypothalamic-pituitary-gonadal axis has been hypothesised, given that luteinising hormone and testosterone levels are suppressed (Figure 2 Left), just as in females with functional hypothalamic oligomenorrhea or amenorrhea (Figure 2 Right). In male athletes, associations between higher levels of intensive endurance training and reductions in testosterone and libido [6,54] have been observed without assessment of EA, and the underlying mechanisms remain unclear. Further research in a controlled setting (i.e., low EA is measured under strict conditions in a laboratory setting) is warranted to understand the causative pathways behind disruption to the male hypothalamic-pituitary-gonadal axis. Furthermore, accurately determining the reliability of testosterone levels as a potential indicator of low EA or RED-S and the impact of low testosterone levels on other testosterone-dependent physiological processes warrants thorough investigation.

## 4. Within-Day Energy Deficiency in Athletes

Recent research into within-day energy balance and within-day energy deficiency, whereby energy intake and exercise energy expenditure are assessed in 1-h intervals, may provide greater insight into real-time changes that are indicative of the endocrine responses associated with variations in EA and may help to identify markers of energy deficiency [55,56]. To date, two studies have examined within-day energy deficiency, one in male [56] and the other in female [55] endurance athletes. In the male study, the relationship between within-day energy deficiency and RMR was explored [56]. Despite similar EA, athletes identified with suppressed RMR spent more time (21 vs. 11 h) with an energy deficit exceeding 400 kcal across a 24-h period. Within-day energy deficiency was associated with higher blood cortisol level and lower testosterone: cortisol ratio [56]. In the female study, within-day energy deficiency was also associated with higher cortisol levels as well as with menstrual dysfunction, lower oestradiol and RMR ratio [55]. Furthermore, despite similar EA, female endurance athletes with menstrual dysfunction spent more time (22 vs. 18 h) in an energy deficient state exceeding 300 kcal compared to eumenorrheic athletes [55]. A positive association between within-day energy deficiency and more frequent meals was also observed. These findings contradict the commonly held belief that higher meal frequency may be necessary for some athletes to attain energy balance and prevent within-day energy deficiency [57]. Consumption of high fibre and low energy-dense foods were previously reported in endurance females with menstrual dysfunction [58]; thus, the energy density of food consumed at mealtimes needs careful consideration to improve within-day energy balance in females with menstrual dysfunction. Other dietary factors such as diet quality and differing attitudes towards food groups need to be investigated when assessing for risk of low EA.

In summary, the viability of measuring within-day energy deficiency and its ability to detect athletes at risk of negative health outcomes associated with low EA is unclear. Questions around appropriate assessment of within-day energy deficiency and the time periods over which it should be assessed impede the quality of research in this area. Further research, especially in female athletes, represents an interesting area for future investigation.

## 5. Low Energy Availability and Sports Performance

While low EA influences many body systems, for example, reproductive system suppression and menstrual cycle disruption as a mechanism to conserve energy, it is important to note other hormonal pathways are altered, resulting in numerous interrelated endocrine-derived physiological consequences [50,59]. These include increased cortisol levels and reduced triiodothyronine (T3), luteinizing hormone (LH) pulsatility and hypoestrogenism [50,59]. Low EA-related menstrual dysfunction is associated with increased bone stress injury risk which can impair training and competition availability [60]. Thus, low EA may be a contributor to poor sports performance due to associated detrimental endocrine effects [61]. A decrease in neuromuscular performance, assessed using isokinetic dynamometry, was observed in elite endurance athletes with menstrual dysfunction in contrast to eumenorrheic endurance athletes [62]. Furthermore, the decreased neuromuscular performance was associated with lower FFM in the leg, glucose, oestrogen, T3, and elevated cortisol [62]. While these findings are unable to provide sufficient evidence of a causal link between these biomarkers and performance, the interrelationship is biologically possible. The study authors hypothesized that a consistently low blood glucose may lead to increased cortisol and reduced T3, in addition to lower muscle mass in the long term, all of which have been associated with reduced neuromuscular performance [62]. Furthermore, these results support previous literature that indicates that physiological manifestations of low EA, such as menstrual dysfunction in female athletes, negatively impact on sporting performance [63,64].

One study, which did not assess EA but instead investigated associations between aspects of endurance exercise training and sexual libido in endurance training males, concluded that libido scores are associated with the duration and intensity of training [6]. These findings indicate that physiological adaptations may occur within males but reliance on self-reported measures limits the interpretation and generalisability of study results [6]. Nonetheless, these preliminary findings indicate that the amount and intensity of endurance training as potential contributing factors to low libido in males needs consideration. Further research is necessary to better understand the impact of low libido on sports performance.

## 6. Knowledge of Low Energy Availability and Relative Energy Deficiency in Sport

Screening and interventions for low EA risk are needed to protect athlete health and performance. Coaches can play a pivotal role in the early identification of athletes at risk of low EA. However, there is little research exploring the knowledge coaches have on this topic. A recent cross-sectional survey that investigated knowledge of the Triad and RED-S among head athletic trainers at National Collegiate Athletic Association (NCAA) member institutions reported that almost all principal athletic trainers (98.6%) were aware of the Triad [10]. Only one-third (33%) were aware of RED-S [10]. Furthermore, 60% and 71% of principal athletic trainers reported that athletes were screened for eating disorders and menstrual irregularities respectively. Those athletes in need of multidisciplinary team support (e.g., athletes identified with menstrual dysfunction or bone stress injury) occurred at division I institutions compared to those in division II or III. A major limitation of this study is that only one third of the principal athletic trainers at NCAA member institutions responded to the survey. It may be that interest and knowledge of the Triad and RED-S was greater in study participants than in those who failed to respond.

Another study investigating risk of low EA and sports nutrition knowledge among female Australian Rules football players reported that 30% of participants were at risk of low EA [8]. Nutrition knowledge of participants was measured using the Sports Nutrition Knowledge Questionnaire [65]. Players answered ~55% of the questions correctly, with the lowest scores observed for responses in the supplement section [8]. Furthermore, an intervention study investigating the effect of a 6-month nutrition education program for male athletes who were at risk of RED-S, reported positive results on bone health and race performance [9]. These findings suggest that nutritional education programs, including specific information on low EA, its associated short and long-term health implications and dietary supplement use, may improve athlete health and performance outcomes. Education on RED-S for athletes and coaches is necessary to encourage screening for and early identification of athletes at risk of low EA. Multidisciplinary healthcare professional input to ensure appropriate interpretation of the results of screening questionnaires and implementation of interventions is required.

A recent intervention investigated changes in Triad knowledge among female high school athletes (*n* = 89) after participation in a number of brief (10-min) Triad educational videos [11]. The educational videos featured (1) a registered dietitian who defined the aetiology and progression of the Triad, and who provided nutritional strategies that may decrease its risk, (2) a former collegiate athlete who shared personal experiences of overcoming Triad-associated problems and, (3) a collegiate coach who shared insights into negative body image and pressure to achieve a specific bodyweight or appearance [11]. Knowledge about consequences of the Triad (for example, menstrual irregularity increases stress fracture risk; low energy intake is negatively associated with menstrual function and bone health) was identified in most participating athletes (≥ 89%) [11]. These findings suggest that educational videos may be an effective method of improving athlete knowledge of the Triad. Whether improved knowledge is associated with behavioural change in athletes with low EA (for example, athletes increasing their energy intake) represents an important area for investigation.

## 7. Conclusions

This article provides a synthesis of the research conducted on RED-S since our previously published review [7]. This is important given the number of studies conducted over the period. Detection of low EA by identifying physiological symptoms and measuring biomarkers associated with the condition has replaced the use of weighed dietary and exercise logs and heart rate monitors or accelerometers. This may facilitate better exploration of the health and performance outcomes associated with low EA. As the majority of studies are cross-sectional in design and cannot show causality, future longitudinal studies with sufficient numbers of those at risk are required to properly understand the consequences of low EA. The poor understanding by athletes and coaches of low EA and its potential health and performance consequences emphasizes the need for further research in this area. Education to increase awareness and to implement dietary and training interventions, in particular, for athletes engaged in intensive training is also vital.

Key points: Until 2015, low energy availability (EA) was predominantly identified using weighed dietary and exercise logs in combination with heart rate monitors or accelerometers [7]. Research published in the past four years has established that low EA is more easily and accurately identified by the use of surrogate markers including suppressed resting RMR [27,38,43] and validated questionnaires that screen for a drive for thinness and physiological symptoms associated with low EA [30]. Understanding athletes’ and coaches’ knowledge and perspectives on low EA and its health and performance consequences is required [10]. Few education initiatives exist to improve athlete understanding of low EA and its associated health and performance consequences [11]. The development and implementation of sports nutrition programmes to increase awareness and improve knowledge of EA and within-day energy balance, as well as sports nutrition treatment strategies for athletes at risk of low EA, are warranted.

## Figures and Tables

**Figure 1 nutrients-12-00835-f001:**
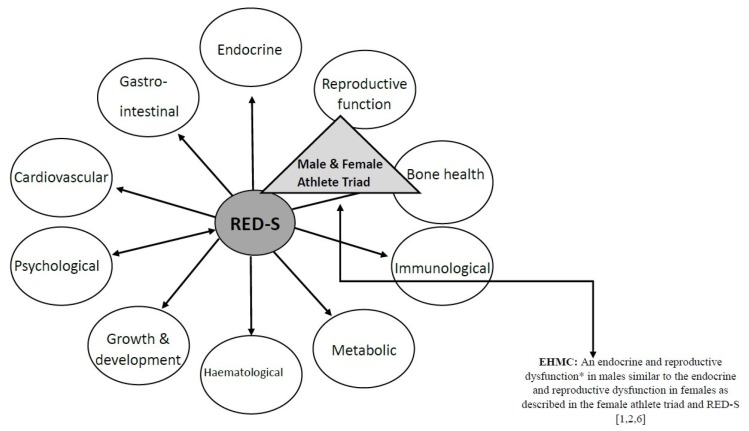
Adapted from the relative energy deficiency in sport health model [1,2] with the inclusion of the male and female athlete triad and the exercise-hypogonadal male condition [6]. **Abbreviations:** EHMC: exercise-hypogonadal male condition; RED-S: relative energy deficiency in sport *the exact physiological mechanism inducing the reduction of testosterone in men is currently unclear; it is postulated to be a dysfunction within the hypothalamic-pituitary-testicular regulatory axis.

**Figure 2 nutrients-12-00835-f002:**
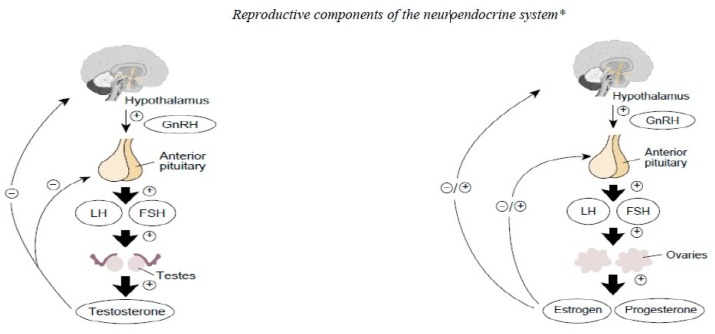
Male and female hypothalamic-pituitary-gonadal axes. Reprinted with permission: Artoria2e5 [CC BY 4.0 (https://creativecommons.org/licenses/by/4.0/) *the reproductive components of the neuroendocrine system in the body are extremely sensitive to Low Energy Availability (LEA) in females [1,2] and the stress of exercise in males [6].

**Table 1 nutrients-12-00835-t001:** Estimated prevalence of low energy availability in various sport groups (Jan 2017–May 2019).

Year	Author	Sex	Sample Size		Athletes	Mean Age (y)	Mean ± SD EA	Subjects with	Comments
							(kcal/kg FFM/Day)*	Low EA* (%)	
Observational Studies									
2019	Civil et al. [14]	F	20		Ballet dancers	18	N/A	22	44% had EA 30–45 kcal/kg FFM/day 22% had EA <30 kcal/kg FFM/day. 40% MD and 65% at risk of LEA. No associations between MD and EA (*p* = 0.17) or LEAF-Q score and EA (*p* = 0.11).



2019	McCormack et al. [25]	M/F	107 27 33 23 24	M F M Ctrl F Ctrl	Cross-country skiers and Ctrl group	20	M:36 ± 16 F:37 ± 21 M Ctrl: 42 ± 15 F Ctrl: 40 ± 21	N/A	F athletes whole-body BMD higher vs. F Ctrl. Higher dietary restraint score in athletes vs. Ctrl. Higher eating concern score in M athletes vs. M Ctrl. Higher shape concern score in F athletes vs. M athletes.
2019	Zabriskie et	F	20		NCAA Division II	20	Off season I: 30 ± 11	N/A	Post hoc comparisons showed that ‘Pre-
	al. [19]				lacrosse athletes		Off season II: 26 ± 11		season’ trended toward a lower EA than in
							Pre-season: 23 ± 9		‘off season I’ (*p* = 0.058) and ‘in season II’
							In season I: 29 ± 10		(*p* = 0.057).
							In season II: 29 ± 9		
2018	Braun et al.	F	56		Elite soccer players	15	N/A	53	Caloric deficit, low carbohydrate and fluid
	[21]								intakes were observed.
2018	Cherian et	M/F	40		Junior national-	12	N/A	M: 24	4 of 5 M and 7 of 11 F with low EA were
	al. [20]		21	M	level soccer players			F: 58	<16 years of age.
			19	F					
2018	Costa et al.	F	21		Collegiate	20	26 ± 13 to 30 ± 13	N/A	Estimated EA was associated with
	[16]				synchronized				measured RMR. No association between
					swimmers				EA and RMR ratio independent of the
									prediction equation used**
2018	Heikura et	M/F	48		Elite distance	M: 27	M:36 ± 6	N/A	No associations between EA and the
	al. [23]		21	M	athletes	F: 26	F:33 ± 7		magnitude of relative change in serum Hb mass.
			27	F					
2018	Heikura et	M/F	59		Elite distance	M: 27	N/A	M: 25	Lower oestradiol, total testosterone, T3 and
	al. [24]		24	M	athletes	F: 26		F: 31	BMD in MD (37%) and low testosterone
			35	F					(40%) athletes.
									Bone injuries: ∼4.5 times more prevalent
									in MD and low testosterone athletes.
2018	Silva et al. [13]	M/F	82 21 M	61 F	Children and adolesc. acrobatic gymnasts	M/F children: 11 M/F adolesc.: 16	CM:54 ± 9 CF:46 ± 9 AM:45 ± 15 AF:33 ± 9	N/A	Lower EA in M and F athletes vs. M and F Ctrl. Most participants did not eat or drink during or immediately after training.



2018	Zanders et	F	13		Collegiate	20	0	N/A	EA did not change across the season.
	al. [17]				basketball players				
2017	Brown et al.	F	25		Pre-professional	21	7-day EA: 26 ± 13	N/A	
	[15]				contemporary		Week EA: 24 ± 10		
					dancers		Weekend EA: 36 ± 21		
2017	Ong et al.	F	9		Dragon boat	23	23.7 ± 13	N/A	Eight of 9 subjects had EA < 45 kcal/kg
	[18]				athletes				FFM/day, with 6 < 30 kcal/kg FFM/day.
2017	Silva et al.	M	151	Rink-hockey***	Children: 10	Children: 48 ± 89	N/A	Lower EI and higher EEE in athletes *vs.*
	[22]		38	Children	players and Ctrl	Adolesc.: 14	Adolesc.: 50 ± 11		Ctrl; resulting in some cases of LEA in
			34	Adolesc.	group		Children Ctrl: 54 ± 9		Athletes.
			43	Children Ctrl			Adolesc. Ctrl: 55 ± 18		
			36	Adolesc. Ctrl					

Abbreviations: Adolesc: adolescents; BMD: bone mineral density; Ctrl: control; EA: energy availability; EEE: exercise energy expenditure; EI: energy intake; F: female; FFM: fat-free mass; Hb; haemoglobin; EA: energy availability; LEAF-Q: low energy availability in females questionnaire; M: male; MD; menstrual dysfunction; N/A: not available; NCAA: national collegiate athletic association; RMR: resting metabolic rate: SD: standard deviation * <30 kcal/kg FFM/day ** prediction equations included the ratio between measured and predicted RMR using the Harris–Benedict equation (HB-RMR ratio), the ratio between measured and predicted RMR using Cunningham equation (C-RMR ratio), the ratio between measured and predicted RMR using different tissue compartments from dual-energy X-ray (DXA) (DXA-RMR ratio) *** similar to ice hockey but played on a dry rink

**Table 2 nutrients-12-00835-t002:** Methods used to assess energy intake and exercise energy expenditure as part of an assessment of energy availability, disordered eating, reproductive function, bone mineral density, body composition and biochemical variables (Jan 2017–May 2019).

Year	Author	Participants	Energy	Exercise	DE	Reproductive	BMD	Body	Biochemical	Other Parameters
		(n)	Intake	Energy		Health		Composition	Parameters	Assessed
				Expenditure					Assessed	
					Methods Used					
Cross-Sectional and Longitudinal Studies										
2019	Civil et al.	20 ballet	Prospective	Accelometer	TFE-Q	Menstrual	DXA	DXA	Vitamin D	Healthier dance
	[14]	dancers	weighed			history				practice national
			dietary			questionnaire				survey
			record			and LEAF-Q				
2019	McCormack	107	FFQ	Activity log	EDE-Q	N/A	DXA	DXA	N/A	N/A
	et al. [25]	27 M runners								
		33 F runners								
		23 M controls								
		24 F controls								
2018	Black et al.	38	Prospective	Activity log	N/A	Menstrual	N/A	Bio-	Serum	N/A
	[33]	recreational	weighed			function		impedance	cholesterols,	
		athletes	dietary			questions in			cortisol,	
			record			the LEAF-Q			progesterone	
									and T3.	
									Salivary	
									testosterone	
2018	Braun et al.	56 F soccer	Prospective	Activity log	N/A	N/A	N/A	Bio-	Serum iron and	N/A
	[21]	players	weighed					impedance	ferritin	
			dietary							
			record							
2018	Cherian et al. [20]	40 soccer players 21 M 19 F	Prospectiveweighed dietary record	HR monitors	N/A	N/A	N/A	4-site skinfold measurements	N/A	N/A
2018	Costa et al. [16]	21 F collegiate synchronized swimmers	Prospective dietary record	Activity log	N/A	N/A	DXA	4- and 7-site	N/A	RMR using indirect
								skinfold		calorimetry
								measurements		
								and DXA		
2019	Zabriskie et	20 NCAA	My Fitness	Accelometer	N/A	N/A	DXA	DXA	N/A	RMR using indirect
	al. [19]	division II	Pal							calorimetry.
		lacrosse	Application							Questionnaire to
		athletes								assess perceived
										rest, soreness and
										training satisfaction
2018	Heikura et	48 elite	Prospective	Activity log	N/A	N/A	N/A	DXA	Serum iron,	Total HB mass
	al. [23]	distance	dietary						ferritin,	
		athletes	record						testosterone and	
		21 M							oestradiol	
		27 F								
2018	Heikura et	59 elite	Prospective	Activity log	N/A	Metabolic and	DXA	DXA	Oestradiol,	Informal
	al. [24]	distance	dietary			reproductive			ferritin, IGF-1,	questionnaire of
		athletes	record			blood			testosterone and	injury and illness
		24 M				hormone			T3	history
		35 F				concentrations				
						and LEAF-Q				
2018	Silva et al.	82 children	Prospective	Activity log	N/A	Menstrual	N/A	3-site	N/A	Sleep duration
	[13]	and adolesc.	dietary			history		skinfold		
		acrobatic	record			questionnaire		measurements		
		gymnasts								
		21 M								
		61 F								
2018	Zanders et	13 F	Prospective	HR monitor	N/A	N/A	DXA	DXA	N/A	RMR using the
	al. [17]	collegiate	dietary	and						Schofield equation,
		basketball	record	accelometer						sleep and recovery
		players								questionnaires
2017	Brown et al.	25 F Pre-	Prospective	Accelometer	TFE-Q	Menstrual	N/A	7-site	N/A	Healthier dance
	[15]	professional	weighed			history		skinfold		practice national
		contemporary	dietary			questionnaire		measurements		survey
		dancers	record and							
			24-h							
			recall							
2017	Ong et al.	9 F Dragon	Prospective	Accelometer	N/A	N/A	N/A	Bio-	N/A	N/A
	[18]	boat athletes	dietary					impedance		
			record							
2017	Silva et al. [22]	72 children and adolesc. M rink- hockey players and 79 M ctrl	Prospective dietary record	Activity log	N/A	N/A	N/A	2-site skinfold measurements	N/A	N/A

Abbreviations: Adolesc: adolescents; BMD: bone mineral density; Ctrl: control; DE: disordered eating; DXA: Dual-energy X-ray absorptiometry; EA: energy availability; EDE-Q: Eating Disorder Examination Questionnaire; F: female; FSH: follicle stimulating hormone; FFQ: food frequency questionnaire; HDL: high density lipoprotein; Hb; haemoglobin; HR: heart rate; IGF-1: insulin-like growth factor; LEAF-Q: Low Energy Availability in Females Questionnaire; LDL: low density lipoprotein; M: male; NCAA: national collegiate athletic association; N/A: not available; NCAA: national collegiate athletic association; RMR: resting metabolic rate; T3: tri-iodothyronine; TC: total cholesterol; TEE: total energy expenditure; TFE-Q: Three Factor Eating Questionnaire; TG: triglycerides.

**Table 3 nutrients-12-00835-t003:** Estimated risk of low energy availability and associated health and performance outcomes.

Year	Author	Sex	Sample Size	Athletes	Mean	% at Risk of Low	% Reporting Health	% Reporting	Comments
					age	EA/Triad/RED-	Outcomes of RED-	Performance	
					(y)	S ^a^	S/Triad	Outcomes of RED-S	
2019	Brook et	M/F	260	Elite para	32	N/A	Prior ED: 3.1	N/A	Most athletes (95 M, 65 F) were
	al. [34]		150 M	athletes			Elevated EDE-Q scores:		attempting to change body
			110 F				32.4		composition/weight to improve
							MD: 44		performance. Athletes with BSI,
							BSI: 9.2		54.5% had low BMD. <10% reported
									awareness of the Triad/RED-S
2019	Condo et al. [8]	F	30	Australian rules	24	30	N/A	N/A	No differences in carbohydrate, protein, fat and energy intakes between those at risk and not at risk of LEA
				football players					

			1000						
2019	Holtzman et al. [35]	F		Adolesc/youngadult athletes	19	Triad risk: 54.7% moderate, 7.9% high; RED-S risk: 63.2% moderate, 33.0% high ^b^	N/A	N/A	The tools agreed on risk for 55.5% of athletes. Agreement ↑ to 64.3% when only athletes with BMD measurements were considered.

2019	Nose- Ogura et al. [36]	F	390	Adolesc/youngadult athletes	21	14 ^c^	MD: 39 Low BMD: 22.7 BSI (last 3 months): 9.2	N/A	Higher BSI risk due to the Triad inteenage athletes vs. athletes in their 20s.
2018	Ackerman et al. [5]	F	1000	Adolesc/young adult athletes	20	47.3 ^d^	MD: 47.9 Impaired bone health: 26.9 Abnormal endocrine function: 3.4 Abnormal metabolic health: 4.4 Impaired haematologicalhealth: 32.1 Impaired growth and development: 14.16 Impaired psychological health: 50.1 Increased cardiovascular risk: 9.5 Impaired GI function: 55.8 Impaired immunological health: 37.5	↓endurance performance: 31.3 ↑injury risk: 38.5 ↓training response: 23.7 Impaired judgement: 8.5 ↓coordination: 20.5 ↓concentration: 14.2 Irritability: 30.7 Depression: 20.7	Increased risk of MD, poor bonehealth, metabolic, haematological and cardiovascular impairment, GI dysfunction, psychological disorders (depression), reduced training response, judgement, coordination, concentration and endurance performance in those at risk vs. not at risk of low EA








2018	Black et	F	38	Recreational	23	63.2	TC > 5.0 mmol/L: 21	N/A	Lower EA, ↓ T3, low energy and
	al. [33]						LDL > 3.0 mmol/L: 25		calcium intake in those at risk of low
									EA
2018	Keay et	M	50	Road cyclists	36	28% ^e^	Lower lumbar spine	N/A	Lack of load-bearing sport associated
	al. [9]						BMD: 44		with low BMD in cyclists with low
									EA. The 10 with low EA had lower
									testosterone levels than those
									maintain adequate EA. Low EA
									associated with reduced body fat percentage.
2018	Logue et	F	833	Elite, sub-elite	N/A	40	≥22 days absence from	N/A	1.7- and 1.8-times increased risk in
	al. [37]			and recreational			training due to illness:		international and provincial/inter-
							24.2		county athletes compared to
									recreationally active individuals
2018	Staal et	M/F	40	Elite ballet	25	F: 40	Low C-RMR: 100 F, 80	N/A	Large variability in suppressed RMR
	al. [38]		20 M	dancers			M		using predictive RMR equations (M:
			20 F				Low HB-RMR ratio: 45		25–80%; F: 35–100%). Cunningham
							F,25M		equation showed highest sensitivity
							Low DXA-RMR ratio: 35		for detecting both genders at risk for
							F,55M		energy deficiency.
2018	Wilson et	M	21	Flat jockeys	A: 19	N/A	N/A	N/A	No difference in RMR or hip and
	al. [39]		17 A		S: 32				lumber spine BMD between groups.
			14 S						Measured RMR did not differ from
									predicted RMR in either group.
2017	Drew et	M/F	132	Elite Olympic	M: 26	40	N/A	N/A	Higher odds of reporting URTI
	al. [40]		47 M	athletes	F: 24				(OR = 3.8), bodily aches (OR = 5.8), GI
			85 F						disturbances (OR = 3.8) and head
									symptoms (OR = 4.4) in those at risk
									of low EA.
2017	Sygo et	F	13	Elite sprinters	21	Pre-training	Pre-training season:	N/A	Primary low EA indicators: a LEAF-
	al. [41]					season: 23 Post-	BMD: 8 RMR: 15 FSH:		Q score >8; RMR < 29 kcal/kg FFM,
						training season:	15		low oestradiol, FSH or LH or a BMD
						39	Post-training season:		of <1.09 g/cm^2^
							BMD: 15 RMR: 8		
							oestradiol: 31 LH: 23		
							FSH: 15		

Abbreviations: A: apprentice flat jockey; Adolesc: adolescents; BEDA-Q: brief eating disorder in athletes questionnaire; BMD: bone mineral density; BP: blood pressure; BSI: bone stress injury; C-RMR ratio: the ratio between measured and predicted RMR using Cunningham equation; DE: disordered eating; DXA-RMR ratio: the ratio between measured and predicted RMR using different tissue compartments from DXA; EA: energy availability; ED: eating disorder; EDE-Q: eating disorder examination questionnaire; EDI-3: eating disorder inventory; ESP: eating disorder screen for primary care; F: female; FFM: fat free mass; FSH: follicle stimulating hormone; GI: gastrointestinal; HB-RMR ratio: the ratio between measured and predicted RMR using the Harris–Benedict equation; LEAF-Q: low energy availability in females questionnaire; LH: luteinising hormone; M: male; MD: menstrual dysfunction; OR: odds ratio; %: percentage; RED-S: relative energy deficiency in sport; RED-S CAT: Relative Energy Deficiency in Sports Clinical Assessment Tool; REDS-outcomes; assessed conditions related to RED-S included blood pressure, eating disorder inventory scores and bone mineral density; RMR: resting metabolic rate; S: senior flat jockey; Triad CRA: Female Athlete Triad Cumulative Risk Assessment ↓ decrease; ↑ increase; URTI: upper respiratory tract infection. ^a^ Risk of LEA/Triad/RED-S assessed using LEAF-Q ^b^ Risk of LEA assessed using the Triad CRA and RED-S CAT ^c^ Risk of LEA defined by body weight ≤85% of the ideal body weight for teenage athletes, or BMI ≤17.5 for athletes in their 20s ^d^ Risk of LEA assessed using the BEDA-Q, ESP and self-reported current or history of ED or DE ^e^ Risk of RED-S assessed using a sport-specific questionnaire and clinical interview (SEAQ-I).

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
