# Peer review of "Low Energy Availability in Athletes 2020: An Updated Narrative Review of Prevalence, Risk, Within-Day Energy Balance, Knowledge, and Impact on Sports Performance"

_nutrients, 2020, doi:10.3390/nu12030835_

Round 1

Reviewer 1 Report

Thank you for allowing me to review this manuscript. It is well written and provides a good update in recent research in the area of low energy availability and RED-s in male and female and where further research is required. It would have been good to highlight how there is limited research on the effect of hormonal contraception on physiological / biochemical assessments of low EA / RED-s and how it is also an area the warrants future research. 

There are a few minor comments which may assist with providing further clarity for the reader as well as future directions of research. 

Abstract: Would change "Development of athlete - and gender" to "Development of sport - and gender..." to clarify if talking about youth, adolescent or senior athletes or athletes in different sports. 

Introduction: The "Key Points" appear better suited as conclusion points rather than in the introduction, considering they do not contain references. 

Following first mention of  International Olympic Committee the abbreviation should be added (IOC), as the abbreviation is then used in the next paragraph. 

3. Low energy availability. 

The first sentence is very long and it gets lost that you are talking about prevalence of Low EA. The various sports are listed in the table and I challenge if they all need to be listed in the text as well. 

It might be this reviewer's ignorance but I am unfamiliar with the term/sport "rink hockey", I would assume it is another way of saying "ice hockey" as it also occurs in a rink, it may be of use to clarify this in the table or abbreviation list. 

The final sentence is again long and verbose with over a line in brackets. Rewording this sentence so that it is shorter would assist with not losing the topic of the sentence and improve flow. 

Figure 2: Please clearly labels "A" and "B" above respective images. 

7, Knowledge of low energy availability...

Following first mention of National Collegiate Athletic Association, the NCAA abbreviation should be added, so it can be used in later in the same paragraph in stead of the full name.  

Conclusion:  Succinct, well written, would benefit from having key points in this section. 

Author Response

Authors’ Response to Reviewers’ Comments

Logue DM et al.

Title: Low energy availability in athletes 2020: An updated narrative review of prevalence, risk, within-day energy balance, knowledge and impact on sports performance

Journal: Nutrients

We would like to take this opportunity to acknowledge and thank the reviewers for their comments and feedback as they have really helped strengthen and improve the quality of the resubmitted manuscript.  The reviewers’ comments are in bold text below.  Our responses are in regular text.  The changes to the revised manuscript are tracked within the manuscript. 

Page numbers cited in this document refer to the revised manuscript.

Authors’ response to Reviewer 1:

General Comments:

Thank you for allowing me to review this manuscript. It is well written and provides a good update in recent research in the area of low energy availability and RED-s in male and female and where further research is required. It would have been good to highlight how there is limited research on the effect of hormonal contraception on physiological / biochemical assessments of low EA / RED-s and how it is also an area the warrants future research. 

Many thanks for this comment.  The above points raised have now been included within the text of the manuscript.

“Nevertheless, further research is warranted as there is limited research on the effect of hormonal contraception on physiological function and biochemical assessment of low EA/RED-S.” (Page 3)

There are a few minor comments which may assist with providing further clarity for the reader as well as future directions of research. 

Abstract: Would change "Development of athlete - and gender" to "Development of sport - and gender..." to clarify if talking about youth, adolescent or senior athletes or athletes in different sports. 

Many thanks for this comment.  This amendment has now been made within the abstract. (Page 1)

“Development of sport and gender-specific screening tools to identify adolescent and senior athletes in different sports at risk of RED-S is warranted.” (Page 3)

Introduction: The "Key Points" appear better suited as conclusion points rather than in the introduction, considering they do not contain references. 

Thank you for this comment.  The key points have now been omitted from the introduction and are now included after the conclusion of the manuscript. (Page 16)

Following first mention of International Olympic Committee the abbreviation should be added (IOC), as the abbreviation is then used in the next paragraph. 

Apologies for this error.  The International Olympic Committee has been abbreviated at first mention and this is included within the introduction of the manuscript. (Page 1). The abbreviation is used subsequently in the manuscript.

  1. Low energy availability. 

The first sentence is very long and it gets lost that you are talking about prevalence of Low EA. The various sports are listed in the table and I challenge if they all need to be listed in the text as well. 

The authors thank the reviewer for this comment and have now changed the long sentence within the manuscript from:

Table 1 summarises the current evidence on the reported prevalence of low EA in a number of sports. Recent studies have investigated the prevalence of low EA in various sports and disciplines including acrobatic gymnasts [9], ballet/contemporary dancers [10, 11], synchronised swimmers [12] and in team sports such as basketball [13], dragon boat racing [14], lacrosse [15], Australian rules football [16], soccer [17, 18] and rink-hockey [19], and ranges from 22 to 58%.

to

Recent studies have investigated the prevalence of low EA in various sports [9-19]. Prevalence ranges from 22 to 58% (Table 1). (Page 2)

It might be this reviewer's ignorance but I am unfamiliar with the term/sport "rink hockey", I would assume it is another way of saying "ice hockey" as it also occurs in a rink, it may be of use to clarify this in the table or abbreviation list. 

The authors have amended the text and no sports are now listed.  Table 1 has been amended to explain the game of rink hockey.

***“similar to ice hockey but played on a dry rink” (Table 1; Page 5).

The final sentence is again long and verbose with over a line in brackets. Rewording this sentence so that it is shorter would assist with not losing the topic of the sentence and improve flow. 

The authors agree and have now changed the sentence from:

“The current evidence implies that clinical, individual assessment of the diagnostic criteria of the Triad [29], the LEAF-Q [28] and the RED-S clinical assessment tool (RED-S CAT) [30] in combination with an evaluation of disordered eating behaviour and reproductive biomarkers (blood biochemical assessment of sex-hormones and prospective collection of oestrogen and progesterone during one or more menstrual cycles) represent a more objective and accurate indicator of optimal EA for health in females than the estimation of EA using dietary and training logs [22].”

to

“The current evidence implies that clinical, individual assessment of the diagnostic criteria of the Triad [29], the LEAF-Q [28] and the RED-S clinical assessment tool (RED-S CAT) [30] is necessary. This assessment should be conducted with an evaluation of disordered eating behaviour and reproductive biomarkers (blood biochemical assessment of sex-hormones and prospective collection of oestrogen and progesterone during one or more menstrual cycles). This may represent a more objective and accurate indicator of optimal EA for health in females than the estimation of EA using dietary and training logs [22]” (Pages 2-3).

Figure 2: Please clearly labels "A" and "B" above respective images. 

Thank you for this recommendation. Figure 2 has now been clearly labelled with A and B above the respective images. (Page 13)  

7, Knowledge of low energy availability...

Following first mention of National Collegiate Athletic Association, the NCAA abbreviation should be added, so it can be used in later in the same paragraph instead of the full name.  

The authors agree with this comment.  The National Collegiate Athletic Association abbreviation has now been included within the text of the manuscript. (Page 14)

National Collegiate Athletic Association (NCAA) member institutions…..

Conclusion:  Succinct, well written, would benefit from having key points in this section. 

Many thanks for this comment.  The key points have now been included after the conclusion of the manuscript. (Page 16)

Reviewer 2 Report

This paper is an excellent review of relative energy deficiency in sport (RED-S).

It is a comprehensive update on the current state of the science that is well-written and logically presented.

The only suggestion I have for the authors is to check the formatting of the Tables - it may just be my "review" copy, but there are several page breaks within each table that interfere with column flow?

Otherwise excellent work.

Author Response

Authors’ Response to Reviewers’ Comments

Logue DM et al.

Title: Low energy availability in athletes 2020: An updated narrative review of prevalence, risk, within-day energy balance, knowledge and impact on sports performance

Journal: Nutrients

We would like to take this opportunity to acknowledge and thank the reviewers for their comments and feedback as they have really helped strengthen and improve the quality of the resubmitted manuscript.  The reviewers’ comments are in bold text below.  Our responses are in regular text.  The changes to the revised manuscript are tracked within the manuscript. 

Page numbers cited in this document refer to the revised manuscript.

Reviewer 2

This paper is an excellent review of relative energy deficiency in sport (RED-S).

It is a comprehensive update on the current state of the science that is well-written and logically presented.

Many thanks for this comment.  No changes to the manuscript are required.

The only suggestion I have for the authors is to check the formatting of the Tables - it may just be my "review" copy, but there are several page breaks within each table that interfere with column flow?

Thank you for this comment and to the journal editorial staff who have corrected this in the updated manuscript version sent to authors.  The authors have amended spacing errors in Table 1 and a column issue in Table 2 (last row within Table 2).  Cells in all three tables have been merged to ensure consistently with line spacing.

Otherwise excellent work.

Thank you for this comment.

Reviewer 3 Report

A current review of this topic is important to uncover in both female and male athletes. Including more depth and breathe of the content provided in this manuscript would better capture the interest of the reader. Some of the paragraphs can be more richly developed by further expanding on the specific topics presented and including more detail and explanation. Here are some comments made to improve the quality of the manuscript as suggested by this reviewer.   

It is important to point out that (at least in women) it is not just that insufficient calories that affects exercise energy expenditure, but that the energy required to sustain normal bodily functions, such as reproductive processes (e.g. menstruation), becomes compromised. That’s why loss of menstrual function or menstrual irregularity is part of the so-called Triad. (you might want to include this in your paper – perhaps in less words).

Key Points – page 1 and 2 – every statement that is bulleted needs to have references attached to each point to support the statements and so that your readers can reference the points brought up on their own and they can find where this information has been reported or published.

Page 2, second paragraph – last sentence should include at the end…..”as the majority of research in the past has focused only on women”.

Paragraph before Methodology – the last sentence: you need to include all your references for this statement (there is mention of published literature, but the reader needs to know the referenced literature).

“The current review aims to highlight”… can you instead say: “The purpose of this investigation is to…….” Please give more explicit reasons why this review was conducted as well as the importance of why it was done. (e.g. why was the review being done and what important information do you hope to gain from conducting the review (what is driving the study?)

Methodology – If articles that reported on awareness and knowledge of RED-S among coaches and athletes were included in the study then this needs to be included in your methodology. Are you able to provide some of your inclusion and exclusion criteria for considering an article for your review? It is stated that the criteria of the “SIGN” was used but many of your readers may not know what these criteria are or may not have access to the reference provided (unless you are restricted by word count and it will make the paper too lengthy).

The authors may want to mention either in the section of LEA on page 3 or at the end of the manuscript in the discussion, the inherent limitations of self-reported questionnaires especially in the population observed and assessed. (It is very likely that they would underreport problems or that they are eating less than they report, for example, given the nature of the condition). Other articles have probably mentioned this but you also want to include it in your paper. 

Page 3: Low energy availability - before this subheading, there should be a heading that says 3. “Results”…then include as subheadings: LEA, LEA risk

For articles Brooke, Holzman, Nose-Ogura, Ackerman and Logue were the subjects included in the study athletes? There is no mention of what sport they may were involved (the title says “performance outcomes in various sporting groups”). Did these articles describe the subject populations, if so, please include. If not, you want to state in your review that they did not report the sport they participating in (you may not want to include it in this table unless you want to exclude the words: “in various sport groups” if there is no mention of the sport the subjects were participating in (please use the word “sport”, not sporting) 

Page 11, 1st paragraph: emphasized (not emphasised)

First paragraph under LEA risk, 1st sentence: Should either be stated as: “The prevalence of low EA……” OR  “The risk of developing….. (not both words: “the prevalence of risk”)

First paragraph under LEA risk: state as “elite para-athletes” (not just “elite para”). Also in this paragraph, it is mentioned that the largest study placed emphasis on body systems beyond……..health”. What are these body systems? Please include the context of what systems the authors described. This information would be of great interest to the reader so please develop this section more by providing more explaining and be an impetus for further discussion. Also, what were the associations – please describe the associations in more specific detail. What kind of “controlled setting” did the authors in the article you reviewed mean? (for example, are these lab based results, or capturing the data during actual performance-based activities (versus those performed in a controlled lab)? There was an association between “low EA risk and illness” – what kind of illness or illnesses did this article describe? Please include more information this section.

Grammar: Moveover, associations have been described (or reported) between….(change the position of the word “described”)

“Those traditionally expected” (can you include some examples, e.g. …..)It is anticipated that potential risk – add potential

Page 13 – last paragraph – research in a controlled setting – what is meant again by this phrase (lab reports, while participating in a physical activity or ?).

Did you have permission to use the Figure that is on page 13? This might be a Figure that is copy-righted by the original authors so you want to get their permission to use it. Same thing for Figures 2 and 2b.

Starting on page 13 – is this the Discussion Section? If so, then your 5, 6, 7 should be subheadings underneath this main section of your paper.

Page 14 (before #6): In summary, the viability – do you mean “feasibility”? Or reliability? Not sure if viability is the correct word to use (its meaning is: “capable of success”) - is the correct meaning for the context of this sentence?

Page 14, Section 6: it would be interesting to expand a little more and summarize the types of injuries were reported and what is meant but other author’s reports of compromised aerobic performance, decreased neuromuscular performance (how were these measurements defined, for example?) and to what degree were the athletes compromised. What is the link between endocrine alterations (menstrual dysfunction) and increased injury risk, reduced muscular strength and compromised aerobic performance. It would important to explain the known or suspected reasons why they are connected or have been observed to occur. The facts are there but the explanations for why they happen (or proposed explanations) would be also be important to include in this paper. There are plenty of associations included but why they happen or are thought to happen would be a nice addition to this section. You can include rationales from other authors but it may also be important to add your own assessment of these studies. (Just an observation: the references at the end of the manuscript are numbered twice). The above revisions are recommended to improve the quality of the manuscript. Thank you for the privilege of reviewing this article.

Author Response

Authors’ Response to Reviewers’ Comments

Logue DM et al.

Title: Low energy availability in athletes 2020: An updated narrative review of prevalence, risk, within-day energy balance, knowledge and impact on sports performance

Journal: Nutrients

We would like to take this opportunity to acknowledge and thank the reviewers for their comments and feedback as they have really helped strengthen and improve the quality of the resubmitted manuscript.  The reviewers’ comments are in bold text below.  Our responses are in regular text.  The changes to the revised manuscript are tracked within the manuscript. 

Page numbers cited in this document refer to the revised manuscript.

Reviewer 3:

A current review of this topic is important to uncover in both female and male athletes. Including more depth and breathe of the content provided in this manuscript would better capture the interest of the reader. Some of the paragraphs can be more richly developed by further expanding on the specific topics presented and including more detail and explanation. Here are some comments made to improve the quality of the manuscript as suggested by this reviewer.   

Many thanks for this comment.  The authors have revised the manuscript to ensure that greater depth and breadth is achieved as recommended in the reviewer’s comments outlined below.  We agree that this improves the overall quality of the manuscript.

Many thanks for this comment.  The authors have revised the manuscript to ensure that greater depth and breadth is achieved as recommended in the reviewer’s comments outlined below.  We agree that this improves the overall quality of the manuscript.

It is important to point out that (at least in women) it is not just that insufficient calories that affects exercise energy expenditure, but that the energy required to sustain normal bodily functions, such as reproductive processes (e.g. menstruation), becomes compromised. That’s why loss of menstrual function or menstrual irregularity is part of the so-called Triad. (you might want to include this in your paper – perhaps in less words).

Thank you for this comment.  The authors agree with this comment and have amended this sentence within the abstract of the manuscript from:

“The condition arises when insufficient calories are consumed to support exercise energy expenditure”

to

“The condition arises when insufficient calories are consumed to support exercise energy expenditure, resulting in compromised physiological processes such as menstrual irregularities in active females.” (Page 1)

Key Points – page 1 and 2 – every statement that is bulleted needs to have references attached to each point to support the statements and so that your readers can reference the points brought up on their own and they can find where this information has been reported or published.

The authors agree and have now referenced each bullet within the key points section. (Page 16)

Key points:

  • Until 2015, low energy availability (EA) was predominantly identified using weighed dietary and exercise logs in combination with heart rate monitors or accelerometers [7].
  • Research published in the past four years has established that low EA is more easily and accurately identified by the use of surrogate markers including suppressed resting RMR [25, 31, 33] and validated questionnaires that screen for a drive for thinness and physiological symptoms associated with low EA [28].
  • Understanding athletes’ and coaches’ knowledge and perspectives on low EA and its health and performance consequences is required [63].
  • Few education initiatives exist to improve athlete understanding of low EA and its associated health and performance consequences [65]. The development and implementation of sports nutrition programmes to increase awareness and improve knowledge of EA and within-day energy balance, as well as sports nutrition treatment strategies for athletes at risk of low EA, are warranted.

Page 2, second paragraph – last sentence should include at the end…..”as the majority of research in the past has focused only on women”.

Thank you for this observation.  The last sentence in the second paragraph on Page 2 has been amended as follows:

“Therefore, the IOC and other researchers in the field (7) advise that research should now focus on the energy demands and performance criteria of males engaged in a range of sports as the majority of research in the past has focused on women [2].”

Paragraph before Methodology – the last sentence: you need to include all your references for this statement (there is mention of published literature, but the reader needs to know the referenced literature).

Thank you for this comment.  The references have now been included as follows:

“Furthermore, the published literature on awareness and knowledge of RED-S among athletes and coaches has been evaluated to identify gaps in the practical application of low EA education in sport [16, 20, 63, 65].” (Page 2)

“The current review aims to highlight”… can you instead say: “The purpose of this investigation is to…….” Please give more explicit reasons why this review was conducted as well as the importance of why it was done. (e.g. why was the review being done and what important information do you hope to gain from conducting the review (what is driving the study?)

The authors would like to thank the reviewer for highlighting this important point.  The text has been amended as follows:

“The current review aims to highlight recent data on the prevalence and risk of low EA, within-day energy deficiency and health and sporting performance consequences associated with the condition.”

to

The purpose of this investigation is to identify other potential methods for assessing low EA given that the current methods used remain challenging. Furthermore, the review highlights recent data on the prevalence and risk of low EA, within-day energy deficiency and the health and sporting performance consequences associated with low EA.” (Page 2)

Methodology – If articles that reported on awareness and knowledge of RED-S among coaches and athletes were included in the study then this needs to be included in your methodology. Are you able to provide some of your inclusion and exclusion criteria for considering an article for your review? It is stated that the criteria of the “SIGN” was used but many of your readers may not know what these criteria are or may not have access to the reference provided (unless you are restricted by word count and it will make the paper too lengthy).

The authors thank the reviewer for these comments and have listed “knowledge and awareness of LEA” in the key search terms.  

The authors apologise for not making the inclusion criteria clear.  The following sentence describing the inclusion criteria has been changed from:

“Only studies that quantified EA by assessing energy intake, exercise energy expenditure (EEE) and body composition and that investigated symptoms associated with low EA within the text of the manuscript were included in this review”

to

“The inclusion criteria was as follows: only studies that quantified EA by assessing energy intake, exercise energy expenditure (EEE) and body composition and that investigated symptoms associated with low EA within the text of the manuscript were included in this review”. (Page 2)

The authors thank the reviewer for the important point made on the SIGN guidelines and have now included the following sentence within the text of the manuscript to describe the criteria:

“The SIGN grading criteria include the assessment of study design and its ability to minimise the possibility of bias as well as an evaluation of the methodological quality, quantity, consistency, and applicability of a study’s results to the evidence base.” (Page 2)

The authors may want to mention either in the section of LEA on page 3 or at the end of the manuscript in the discussion, the inherent limitations of self-reported questionnaires especially in the population observed and assessed. (It is very likely that they would underreport problems or that they are eating less than they report, for example, given the nature of the condition). Other articles have probably mentioned this but you also want to include it in your paper. 

The authors agree with this comment and have now included the following at the end of section 3.2.1 to highlight the limitations of using self-reported questionnaires in athlete populations.

“Nevertheless, self-reported data need to be interpreted with caution as they may be heavily influenced by athletes’ perceptions, experiences and recollections, thus, further emphasising the need to use well validated questionnaires to investigate outcomes associated with LEA in the athletic population.” (Page 3)

Page 3: Low energy availability - before this subheading, there should be a heading that says 3. “Results”…then include as subheadings: LEA, LEA risk

The authors have now included a subheading “3. Results” and the subheadings thereafter have been changed to “3.1 low energy availability, 3.2 LEA risk etc…..”.

For articles Brooke, Holzman, Nose-Ogura, Ackerman and Logue were the subjects included in the study athletes? There is no mention of what sport they may were involved (the title says “performance outcomes in various sporting groups”). Did these articles describe the subject populations, if so, please include. If not, you want to state in your review that they did not report the sport they participating in (you may not want to include it in this table unless you want to exclude the words: “in various sport groups” if there is no mention of the sport the subjects were participating in (please use the word “sport”, not sporting) 

The authors thank the reviewer for this observation.  The studies listed above stated that female athlete populations were investigated.  However, the various sporting groups were not obvious from the text.  The authors agree with the suggestion made to amend the wording of the title of Table 3.  For clarity, the following words have been removed from the title of Table 3: “in various sporting groups”. 

Page 11, 1st paragraph: emphasized (not emphasised)

Apologies for this spelling error.  The word is now correctly spelled within the text. (Page 12)

First paragraph under LEA risk, 1st sentence: Should either be stated as: “The prevalence of low EA……” OR  “The risk of developing….. (not both words: “the prevalence of risk”)

Thank you for this observation. The sentence has been amended as follows:

“The risk of low EA has been investigated…” (Page 12)

First paragraph under LEA risk: state as “elite para-athletes” (not just “elite para”).

This amendment has been made within the text of the manuscript. (Page 12)

Also in this paragraph, it is mentioned that the largest study placed emphasis on body systems beyond……..health”. What are these body systems? Please include the context of what systems the authors described. This information would be of great interest to the reader so please develop this section more by providing more explaining and be an impetus for further discussion. Also, what were the associations – please describe the associations in more specific detail. What kind of “controlled setting” did the authors in the article you reviewed mean? (for example, are these lab based results, or capturing the data during actual performance-based activities (versus those performed in a controlled lab)?

The body systems (listed below) and the context of the body systems described by the authors have been included as follows:

“The largest cross-sectional study investigating self-reported health and performance outcomes linked to low EA and RED-S [5] placed emphasis on including body systems beyond reproductive function and bone health such as metabolic, haematological, psychological and cardiovascular health and gastrointestinal function.” (Page 12)

The authors included associations between risk of low EA and health and performance outcomes in Table 3. Reference to these associations is now included within the text of the manuscript (see wording below, Page 12). The rationale for describing the associations within Table 3 was to avoid exceeding the manuscript word count specified by the Journal. The need to investigate associations within a controlled setting is now emphasized within the text as follows:

“The authors acknowledged that associations identified in their study (Table 3) were based on self-reported data and highlight the need to investigate the health and performance components of RED-S in a controlled setting, whereby low EA is measured under strict conditions in a laboratory setting, to understand causative pathways [5].” (Page 12)

There was an association between “low EA risk and illness” – what kind of illness or illnesses did this article describe? Please include more information this section.

The authors of the study reviewed classified an illness case as: self-reporting missing training for a period of 24 or more consecutive hours due to medical illness in the month prior to the completion of the questionnaire.  Data on the type of illness are not available.  We have updated this sentence to the following:

“Another cross-sectional study supports associations between low EA risk and self-reported medical illness [37].” (Page 12)

Grammar: Moveover, associations have been described (or reported) between….(change the position of the word “described”)

This has now been changed within the text of the manuscript:

“Moreover, associations have been described between low EA risk and stress fractures, absence from training for > 22 days due to illness and reported adherence to a gluten-free diet [41].” (Page 12)

“Those traditionally expected” (can you include some examples, e.g. …..)It is anticipated that potential risk – add potential

Examples of symptoms traditionally expected have now been included within the text of the manuscript:

“These findings suggest that those at risk of low EA can present with symptoms other than those traditionally expected e.g. menstrual irregularities, and highlight the complexity of identifying individuals at risk [37].” (Page 12)

“Potential” has now been added to the following sentence:

“Following its development and validation, it is anticipated that potential risk of low EA will be identified in a range of athletic male populations. (Page 12)

Page 13–last paragraph – research in a controlled setting – what is meant again by this phrase (lab reports, while participating in a physical activity or ?).

Thank you for this comment and observation.  This sentence has now been amended within the text as follows:

“Further research in a controlled setting (i.e. low EA is measured under strict conditions in a laboratory setting) is warranted to understand the causative pathways behind disruption to the male hypothalamic-pituitary-gonadal axis.” (Page 14)

Did you have permission to use the Figure that is on page 13? This might be a Figure that is copy-righted by the original authors so you want to get their permission to use it. Same thing for Figures 2 and 2b.

Thank you for these important observations.

Figure 1 was adapted by the authors.  The references from the publications that used the figures from which our figure has been adapted are included.

Figure 2A and Figure 2B are Creative Commons images available in the public domain.  The citation- “Reprinted with permission: Artoria2e5 [CC BY 4.0]” has been included within the text of the manuscript to ensure that these are appropriate referenced.

Starting on page 13 – is this the Discussion Section? If so, then your 5, 6, 7 should be subheadings underneath this main section of your paper.

Yes, this is correct.  Subheadings have been corrected within the manuscript.

Page 14 (before #6): In summary, the viability – do you mean “feasibility”? Or reliability? Not sure if viability is the correct word to use (its meaning is: “capable of success”) - is the correct meaning for the context of this sentence?

The authors agree with this comment and have now amended the wording as follows:

“Furthermore, accurately determining the reliability of testosterone levels as a potential indicator of low EA or RED-S and the impact of low testosterone levels on other testosterone-dependent physiological processes warrants thorough investigation.” (Page 14)

Page 14, Section 6: it would be interesting to expand a little more and summarize the types of injuries were reported and what is meant but other author’s reports of compromised aerobic performance, decreased neuromuscular performance (how were these measurements defined, for example?) and to what degree were the athletes compromised. What is the link between endocrine alterations (menstrual dysfunction) and increased injury risk, reduced muscular strength and compromised aerobic performance. It would important to explain the known or suspected reasons why they are connected or have been observed to occur. The facts are there but the explanations for why they happen (or proposed explanations) would be also be important to include in this paper. There are plenty of associations included but why they happen or are thought to happen would be a nice addition to this section. You can include rationales from other authors but it may also be important to add your own assessment of these studies. (Just an observation: the references at the end of the manuscript are numbered twice). The above revisions are recommended to improve the quality of the manuscript. Thank you for the privilege of reviewing this article.

Many thanks for these comments.  The authors fully agree that more detail and clarity is needed around the studies included. This paragraph has been rewritten to improve the overall quality of our review.

The paragraph now reads as follows:

“While low EA influences many body systems, for example, reproductive system suppression and menstrual cycle disruption as a mechanism to conserve energy, it is important to note other hormonal pathways are altered, resulting in numerous interrelated endocrine-derived physiological consequences [48, 57]. These include increased cortisol levels and reduced triiodothyronine (T3), luteinizing hormone (LH) pulsatility and hypoestrogenism [48, 57]. Low EA-related menstrual dysfunction is associated with increased bone stress injury risk which can impair training and competition availability [58]. Thus, low EA may be a contributor to poor sports performance due to associated detrimental endocrine effects [59]. A decrease in neuromuscular performance, assessed using isokinetic dynamometry, was observed in elite endurance athletes with menstrual dysfunction in contrast to eumenorrheic endurance athletes [60]. Furthermore, the decreased neuromuscular performance was associated with lower FFM in the leg, glucose, oestrogen, T3, and elevated cortisol [60]. While these findings are unable to provide sufficient evidence of a causal link between these biomarkers and performance, the interrelationship is biologically possible. The study authors hypothesized that a consistently low blood glucose may lead to increased cortisol and reduced T3, in addition to lower muscle mass in the long term, all of which have been associated with reduced neuromuscular performance [60]. Furthermore, these results support previous literature that indicates that physiological manifestations of low EA, such as menstrual dysfunction in female athletes, negatively impact on sporting performance [61, 62].”

Apologies for the duplication of reference numbers.  This has now been corrected.